# Er:YAG Laser Irradiation Reduces Microbial Viability When Used in Combination with Irrigation with Sodium Hypochlorite, Chlorhexidine, and Hydrogen Peroxide

**DOI:** 10.3390/microorganisms7120612

**Published:** 2019-11-25

**Authors:** Janina Golob Deeb, John Smith, B. Ross Belvin, Janina Lewis, Kinga Grzech-Leśniak

**Affiliations:** 1School of Dentistry, Virginia Commonwealth University, Richmond, VA 23298, USA; jgolobdeeb@vcu.edu (J.G.D.); smithjl47@mymail.vcu.edu (J.S.); 2Philips Institute, Virginia Commonwealth University, Richmond, VA 23298, USA; belvinbr@vcu.edu (B.R.B.); jplewis@vcu.edu (J.L.); 3Department of Oral Surgery, Wroclaw Medical University, 50-425 Wroclaw, Poland

**Keywords:** Er:YAG laser, *Fusobacterium nucleatum*, hydrogen peroxide, laser, oral pathogens, periodontitis, *Porphyromonas gingivalis*, sodium hypochlorite, *Streptococcus gordonii*, chlorhexidine

## Abstract

The erbium-doped yttrium aluminum garnet (Er:YAG) laser is used to treat periodontal disease; however, its effectiveness at killing oral bacteria is not well known. Furthermore, the compounding effect of the combination of a laser treatment and irrigation methods with antimicrobials on bacterial viability is yet to be determined. The purpose of this in vitro study was to evaluate the effect of the Er:YAG laser with irrigation using chlorhexidine (CHX), hydrogen peroxide (H_2_O_2_), or sodium hypochlorite (NaOCl) on the viability of oral bacteria. Three bacterial species were used in our study: *Streptococcus gordonii*, *Fusobacterium nucleatum*, and *Porphyromonas gingivalis*. Bacteria were grown in an anaerobic chamber in brain heart infusion broth and incubated at 37 °C. Bacterial samples with an OD of 0.5 were irradiated with the Er:YAG laser at 2940 nm using a 400-µm Varian tip. The experiment was repeated four times using these parameters: 40 mJ, 40 Hz, and 1.6 W for 20 seconds with the 300 µs short pulse duration in contact mode. Treatment groups consisted of the following: (1) no treatment, (2) 0.5% H_2_O_2_ alone, (3) 0.5% NaOCl alone, (4) 0.03% CHX alone, (5) Er:YAG irradiation alone, (6) Er:YAG irradiation with 0.5% H_2_O_2_, (7) Er:YAG irradiation with 0.5% NaOCl, and (8) Er:YAG irradiation with 0.03% CHX. Microbial viability was determined through plating and colony counts and calculated into CFU/ml. Statistical analysis was done using a two-tailed paired t-test. The use of the Er:YAG laser alone failed to show statistically significant antibacterial activity against any of bacteria. The most effective mono-treatment with irrigation solutions for all three bacteria were 0.5% H_2_O_2_ and 0.5% NaOCl (*p* < 0.001 for each solution). Irrigation with 0.03% CHX was most effective against *F. nucleatum* (*p* < 0.01) and less against *P. gingivalis* and *S. gordonii* and showed the least antibacterial action alone but improved significantly in combination therapy (*p* < 0.05). The combined treatment with the Er:YAG showed the greatest and most significant improvement in the reduction of bacterial viability compared to any other treatment group (*p* < 0.05 for each combined treatment). Irradiation with the Er:YAG laser with the addition of 0.5% H_2_O_2_, 0.5% NaOCl, or 0.03% CHX under a short working time (20 s) resulted in a significant reduction of bacterial viability for all three bacterial species compared with any single treatment option. The combination of irradiation with the Er:YAG laser with the addition of 0.5% H_2_O_2_, 0.5% NaOCl, or 0.03% CHX resulted in a larger reduction of bacterial survival when compared to monotherapies with antimicrobial solutions or laser. The combination of the Er:YAG laser with a low concentration irrigant solution of 0.5% H_2_O_2_, 0.5% NaOCl, or 0.03% CHX could be an effective treatment protocol for the reduction of periodontal pathogens and thus suitable treatment for non-surgical periodontal therapy.

## 1. Introduction

Periodontitis is a chronic, multifactorial, polymicrobial inflammatory condition initiated by bacteria in dental plaque biofilms, eventually leading to inflammation and bone loss [1]. It is one of the most prevalent diseases worldwide and, in advanced cases, the major cause of tooth loss in adults [2]. Among the bacteria implicated in periodontal disease are mainly anaerobic ones belonging to the red complex such as *Porphyromonas gingivalis* and *Tannerella forsythia* [3]. Bacteria belonging to the orange complex such as *Fusobacterium nucleatum* and *Prevotella intermedia* provide an attachment bridge for several pathogenic bacteria as well as reduce oxygen tension, thus providing a hospitable environment for the development of bacterial complexity in dental plaque biofilms [3,4,5]. Early colonizers, such as *Streptococcus gordonii*, are the ones associated with the healthy state of the host and provide attachment sites for later colonizers [6,7]. Recent studies show that periodontitis is a polymicrobial condition, rather than a disease resulting from monoinfection [1,8]. Periodontal pathogenic microbes and their endotoxins are considered a primary etiologic factor. They adhere and infiltrate the infected root surface cementum and pocket epithelium and remain in the surrounding connective tissue and bone [9]. Periodontal therapy aims to reduce periopathogenic bacteria by mechanical disruption, which can be complemented with antibacterial therapy including local and systemic antibiotics [10]. The frequent use of antibiotics has been contributing to a global increase in bacterial resistance and undesirable side effects, which can have a negative impact on health and treatment outcomes [11,12]. Alternative therapies have been explored, including the use of lasers and chemotherapeutic agents [13,14,15,16].

Lasers have been used for decades in various fields of dentistry, including periodontal and peri-implant therapy [17,18,19,20,21,22,23,24]. High-power lasers such as the erbium-doped yttrium aluminum garnet (Er:YAG) with a wavelength of 2940 nm are especially useful for ablation, vaporization, and disinfection, and also demonstrate beneficial biological effects, including the promotion of tissue regeneration, photobiomodulation, enhanced wound healing, and overall improved clinical outcomes on the host [16,23]. It is noteworthy that the Er:YAG laser has the highest absorption in water and could be a suitable treatment modality for the reduction of bacteria associated with periodontal disease. This laser, when used with scaling and root planning, leads to a significant improvement in periodontal inflammation and decreased probing depths [25,26,27]. Most studies investigating the antimicrobial effectiveness of the Er:YAG laser used either scanning microscopy or the detection of bacteria based on the levels of DNA; thus, data directly investigating the effectiveness of the Er:YAG laser on bacterial viability are lacking [28,29,30].

Only a small number of studies has explored the effectiveness of laser irradiation in combination with antiseptic therapy [31,32]. Findings from those studies suggest that laser irradiation with H_2_O_2_ photolysis in conjunction with mechanical therapy for the treatment of periodontitis is an effective non-surgical treatment modality resulting in a reduction of pocket depth and counts of *P. gingivalis* [32]. Other antiseptics such as chlorhexidine (CHX) digluconate, hydrogen peroxide (H_2_O_2_), and sodium hypochlorite (NaOCl) are yet to be tested in combination therapy with an Er:YAG laser. At 0.2% concentration, CHX showed effective inhibition in vitro against *P. gingivalis* and *F. nucleatum* [33]. The use of 3% H_2_O_2_ resulted in a reducing pocket of more than 4 mm, with no effect on bleeding and other gingival indexes [34]. Sodium hypochlorite combined with curettage effectively reduced soft tissue inflammation [35], as well as improved healing and the regeneration of the connective tissue [36].

The concept of combining antiseptic agents with Er:YAG laser irradiation and studying the effect on bacterial viability is yet to be investigated more comprehensively. The purpose of this in vitro study was to evaluate the effectiveness of the combination therapy of Er:YAG laser irradiation with low concentrations of H_2_O_2_, NaOCl, and CHX solutions on three oral bacteria representing the different functions within different complexes the diseased biofilm: *S. gordonii* (primary colonizer, health promoting), *P. gingivalis* (red complex, pathogenic bacterium), and *F. nucleatum* (bridging bacterium). We hypothesized that a combination treatment would generate an additive bactericidal effect compared to laser or chemical irrigation alone. 

## 2. Material and Methods

### 2.1. Culture Conditions

Three oral bacterial species implicated in the colonization of the periodontal pocket and progression of periodontal disease were used in this study: *P. gingivalis*, *Pg* (W83), *F. nucleatum*, *Fn* (ATCC 25566), and *S. gordonii*, *Sg* (ATCC 10558). They were individually grown but treated in parallel. Freezer stock (−80 °C) of the bacterial species were streaked onto blood agar plates (TSA II, 5 % sheep blood; BBL) and incubated at 37 °C in an anaerobic chamber consisting of 10% CO_2_, 10% H_2_, and 80% N_2_ for 48 hours. A single colony-forming unit (CFU) was used to inoculate 3 ml of brain heart infusion (BHI) broth. The inoculum was incubated overnight in an anaerobic environment at 37 °C. The optical density (OD) of the cultures was measured with a spectrophotometer at 660 nm (OD660) and normalized to an OD of 0.5. Then, these cultures were aliquoted into a 96-well plate. Thus, 75 μL of each of the cultures was aliquoted to 16 wells/bacterial strain of the 96-well plate to facilitate experiments done with eight treatment groups in duplicate. For treatment, chemicals were added in the form of concentrated stock solutions (H_2_O_2_: 30%, NaOCl: 10%, CHX: 0.12%) directly to the bacteria cultures in the plate at concentrations specified per study group. After the addition of chemical treatments, the 96-well plate was removed from the anaerobic chamber and placed into a sterile biological safety hood. To check the effect of aerobiosis on growth of bacteria, the control cells were moved to the anaerobic chamber without treatment. Laser irradiation was performed on the appropriate samples aerobically. Following laser irradiation, the 96-well plate was placed immediately back into the anaerobic chamber where each treated well was diluted 1:10 into fresh BHI broth. Additional dilutions were performed and inoculated onto blood agar plates. The plates were incubated for 2–7 days anaerobically at 37 °C. Viable colonies were counted for each plate, calculated into CFU/ml, and converted into log form for statistical analysis.

### 2.2. Laser Irradiation PARAMETERS

The Er:YAG laser was set to normal periodontal clinical settings with a short working time of 50 seconds. The samples were irradiated by Er:YAG laser at 2940 nm (LightWalker, Fotona, Slovenia), using a 400-µm Varian fiber tip of cylindrical quartz at parameters: 40 mJ; 40 Hz; 1.6 W for 20 seconds, with the 300 µs short pulse duration in contact mode. Irradiation was performed with a disinfected aluminum foil barrier to isolate treated wells from contamination.

### 2.3. Study Groups

Group 1: bacteria alone (*Fn* alone or *Pg* alone or *Sg* alone)

Group 2: bacteria (*Fn* alone or *Pg* alone or *Sg* alone) + H_2_O_2_ (0.5%)

Group 3: bacteria (*Fn* alone or *Pg* alone or *Sg* alone) + NaOCl (0.5%)

Group 4: bacteria (*Fn* alone or *Pg* alone or *Sg* alone) + CHX (0.03%)

Group 5: bacteria (*Fn* alone or *Pg* alone or *Sg* alone) + laser Er:YAG alone

Group 6: bacteria (*Fn* alone or *Pg* alone or *Sg* alone) + laser Er:YAG + H_2_O_2_ (0.5%)

Group 7: bacteria *(Fn* alone or *Pg* alone or *Sg* alone) + laser Er:YAG + NaOCl (0.5%)

Group 8: bacteria (*Fn* alone or *Pg* alone or *Sg* alone) + laser Er:YAG + CHX (0.03%)

Each experiment was performed in duplicate for technical replicates. The experiment was done at least on four different days, thus giving four biological replicates. 

### 2.4. Statistical Analysis

The survival rates of bacterial species were statistically analyzed separate from one another, with comparisons amongst treatment groups for the same bacterium. The significance between each group was calculated using a two-tailed paired *t*-test. Changes were considered significant at the *p* < 0.05 = *, *p* < 0.01=**, and *p* < 0.001 = *** level. 

## 3. Results

The application of diluted antimicrobial irrigants in combination with an Er:YAG laser produced an environment that greatly reduced bacterial viability in a short working time (20 s). Combined treatment with the Er:YAG laser and irrigation solutions using 0.5% NaOCl, 0.5% H_2_O_2_, or 0.03% CHX resulted in the most effective reduction of bacteria in all three bacterial species compared with any single treatment option. The combination treatment in some groups decreased the bacterial viability to the limit of detection.

The results of single and combined treatments against *P. gingivalis* are shown in Table 1, and in Figure 1 and Figure 4. We found that *P. gingivalis* had a higher sensitivity to antimicrobial irrigation alone when compared to the other two bacterial species. The most effective mono-treatment with irrigation solutions for *P. gingivalis* were 0.5% H_2_O_2_ and 0.5% NaOCl (highly statistically significant with *p* < 0.001 for each solution). Treatment with 0.03% CHX showed a statistically significant diminished bacterial load (*p* < 0.01); however, the average bacterial load was decreased by only 1.73 Log (CFU/mL). The use of Er:YAG laser alone failed to show statistically significant antibacterial activity against *P. gingivalis*, but the combined (laser/irrigation) treatment showed further strengthening of antimicrobial action of irrigation solutions. The difference was not statistically significant for existing strong antibacterial 0.5% H_2_O_2_ and 0.5% NaOCl monotherapies (*p* > 0.05 each), but a weak antibacterial effect of 0.03% CHX was synergistically pushed well into the range of statistical significance (*p* < 0.01) (Table 1, Figure 1).

The effects of treatments for *F. nucleatum* are shown in Table 1 and Figure 2 and Figure 4. Similarly to *P. gingivalis*, treatment with 0.5% H_2_O_2_ and 0.5% NaOCl in solo irrigation treatment had strong antibacterial effect (*p* < 0.001 for each solution). Irrigation with 0.03% CHX alone also showed statistically significant antibacterial action (*p* < 0.01). The overall bacterial load drop reached an average of 5.71 Log (CFU/mL), which is a strong effect when compared between bacterial species (for *S. gordonii*, 1.94 logs; for *P. gingivalis*, 1.73 logs; both on average). The laser alone did not significantly lower the bacterial viability, but it did improve the antibacterial effects of irrigants in combined therapy. Although this finding was not significant for 0.03% CHX (*p* > 0.05), the combined therapy of Er:YAG with 0.5% H_2_O_2_ and 0.5% NaOCl notably lowered the number of bacteria below the limits of detection for each of the combined treatments (Figure 2).

*S. gordonii* is aerotolerant bacterium and was more resistant to chemical treatments when compared to *P. gingivalis* and *F. nucleatum*. The results for *S. gordonii* are presented in Table 1 and Figure 3 and Figure 4. Although all irrigation solutions had statistically significant antibacterial action in monotherapy (*p* < 0.001 each), the combined treatment with the Er:YAG laser showed the greatest and most significant improvement in the reduction of bacterial viability compared to any other treatment group (*p* < 0.05 for each combined treatment). Again, 0.03% CHX showed the least antibacterial action alone but improved significantly in combination with laser therapy (*p* < 0.05) (Figure 3 and Figure 4).

In summary, the results indicate that the Er:YAG laser alone had limited antibacterial effect, but in combination with 0.5% H_2_O_2_ or 0.5% NaOCl, it showed a notable reduction in bacterial viability regardless of the bacteria tested. The addition of laser irradiation to each of the irrigation treatment modalities resulted in a statistically significant reduction of bacteria as compared to the use of antibacterial solutions alone for *S. gordonii.* For *P. gingivalis,* the statistically significant decrease in bacterial viability following the addition of the laser was found for the CHX solution only. A visible decrease in bacterial viability was noted for the other study groups following the addition of laser irradiation. For *F. nucleatum,* the combination of Er:YAG laser irradiation and 0.5% H_2_O_2_ or 0.5% NaOCl reduced the levels of viable bacteria below the limit of detection. 

## 4. Discussion

Our study shows that Er:YAG laser treatment with the addition of three antiseptic agents at low concentrations is an effective antimicrobial treatment. In clinical practice, the Er:YAG laser and antiseptic solutions are used as adjunct to mechanical periodontal therapy. Human histological studies have shown that treatment with an Er:YAG laser results in favorable healing, the formation of new attachment, and may result in radiographic bone regeneration [37]. The mechanism of action of the Er:YAG laser is based on the absorption of its energy in the water and organic molecules within the cells resulting in the vaporization of these molecules within the tissues and increasing intra-tissue pressure by producing vapor and provoking “micro-explosions” that cause mechanical breakdown [13,15,38].

Irradiation with the Er:YAG laser facilitates root debridement and has positive effects on tissue repair, resulting in greater probing depth reduction and gains in clinical attachment level that are sustained for up to three years compared to flap surgery [39]. Various settings have been proposed for the clinical use of the Er:YAG laser for treatment of periodontal disease [25,27,28,29,40]. The working parameters used in this study are safe and consistent with previously reported data [25,27]. A combination therapy of laser irradiation at normal clinical working parameters in combination with antiseptic agents at low concentrations shows promising results in the reduction of bacterial viability when compared to any of the monotherapies, which represent the most frequently applied treatments in the clinical practice. These observations suggest that laser-activated solution using 0.5% NaOCl or 0.5% H_2_O_2_ combination treatment generated an additive bactericidal effect resulting in a more effective reduction of bacterial viability when compared to the laser irradiation or chemical irrigation alone. A 0.1–0.5% solution of NaOCl has been proposed as an antiseptic oral irrigation due to its rapid bactericidal action, relative non-toxicity at suggested concentrations, no color, no staining, very low cost, and no known contraindications [41]. Thus, it is a promising candidate for use in combination therapy. At the concentrations used, CHX had the lowest effect in reducing microbial viability when compared to the other antimicrobial solutions tested, which is a surprising observation considering that it is the most frequently used oral antiseptic in periodontal clinical practice. More research should be done to abate any potential for confounding variables affecting these results^.^

Photolysis at even higher concentrations of 3% H_2_O_2_ on an animal model simulating clinical conditions resulted in no mucosal irritation or abnormal findings macroscopically or microscopically. Hydroxyl radicals released from H_2_O_2_ by photolysis are powerful oxidizing agents that are capable of inducing lethal oxidative damage to microbes [31]. The lack of residual action of these radicals at the treated sites may be due to their extremely short half-life in liquid [42].

Irradiation by most surgical lasers exerts photo-thermal effects on target tissues and cells that are capable of killing bacteria by evaporation, destruction, and denaturation, resulting in their devitalization or inactivation [43,44,45,46,47,48]. The bactericidal effects of laser therapy are considered capable of creating a disinfected field and reducing the risk of infection [49]. Hard-tissue lasers such as Er:YAG are the most promising and widely employed for root-surface and bone-defect debridement during surgery [13,22,38] with the capability of extensive bacterial eradication into deeper layers of dentin at no significant temperature increase [48]. More complex biological effects include biostimulation, which may help accelerate wound healing and tissue regeneration [16,17,50]. Although efficient when used alone, the Er:YAG laser may not be more effective for subgingival debridement than conventional ultrasonic instrumentation in changing the microbial composition of pathogens or probing depth [51]. However, when used as a combined therapy, such treatment shows promise for clinical settings. Bacterial species in subgingival plaque exist in complexes that are inter-related, suggesting that therapies affecting some species may influence the colonization of others. The treatments in this experiment disrupted late and intermediate colonizers more than early colonizers, which could enable the reduction of periopathogenic anaerobic Gram-negative bacteria in favor of aerobic Gram-positive bacteria, facilitating a shift to a healthier microbial composition. Combined laser therapy could effectively target multiple bacteria associated with periodontitis and enhance the effects of periodontal treatment, since the long-term stability of periodontal attachment depends on the reduction of bacterial deposits on the root surface and plaque control [52]. More work in this area is yet to be done. Future studies should be designed to explore whether these therapies have the same effects on a more complex microbial biofilm as they do on individual bacteria. Potential interactions among different bacteria and the use of a more complex biofilm would bring a model closer to oral cavity. 

This study was not without limitations. We intended to determine only the effect on bacterial viability, and thus no host factors were considered. Due to the physical size of the laser, bacterial cultures had to be moved from the anaerobic environment for treatment purposes. All treatment groups were handled identically in that process and thus can be compared among themselves. In addition, all bacterial species were shown to be able to withstand oxygen exposure, which further reduces the significance of the oxygen exposure in this study. Our study only examined three bacterial species, and thus further studies are needed studying more complex microbial mixtures to determine the effectiveness of the combined therapy. Some treatment groups resulted in no detectable colonies, making comparisons more difficult. However, such data underscored the effectiveness of the treatment. 

The findings of this study could be easily and safely applied to clinical use, as these treatment modalities already exist in clinical practice as monotherapies. Combination therapies of an Er:YAG laser and a dilute oxidative solution appear to be more effective and could be implemented into non-surgical periodontal therapy as a non-invasive treatment facilitating a decrease in the bacterial viability. The Er:YAG laser has also been used for treatment of peri-implantitis with similar exposure settings [20,53]. Clinical applications should be explored and studied to examine our in vitro findings and study the efficiency of these protocols in treatment of periodontitis and peri-implantitis. 

## 5. Conclusions

This is the first study demonstrating that the Er:YAG laser irradiation with the energy of 40 mJ, 40 Hz, and 1.6W for 20 seconds together with a very low concentration of 0.5% hydrogen peroxide or 0.5% sodium hypochlorite provides superior results in antibacterial effects on oral bacteria than the same treatments delivered as monotherapies. The most clinically relevant finding is that the use of laser treatment in combined therapy with a weak irrigation solution has a stronger effect than each treatment as monotherapy, and the common implementation of such combined modality may improve treatment outcomes. 

## Figures and Tables

**Figure 1 microorganisms-07-00612-f001:**
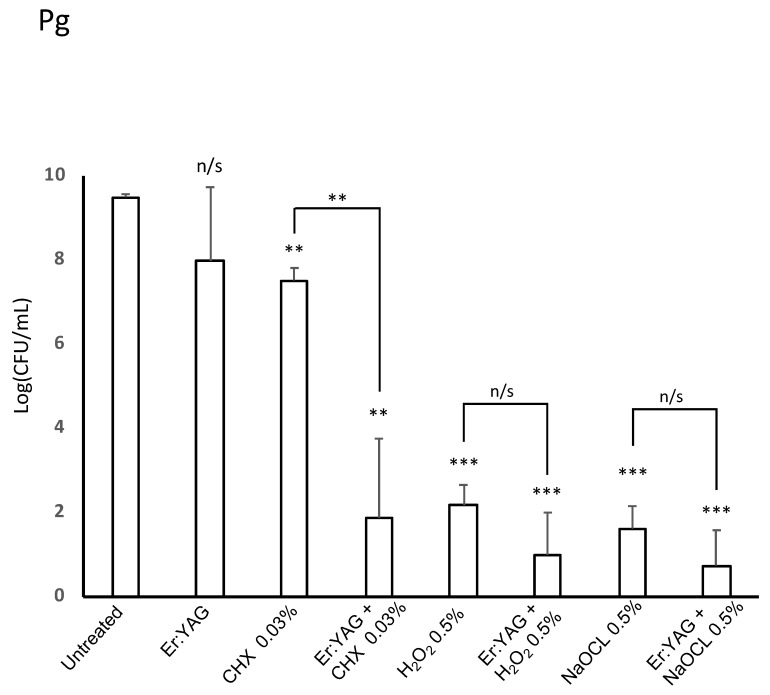
Effectiveness of erbium-doped yttrium aluminum garnet (Er:YAG) laser treatment with or without irrigation on the viability of *P. gingivalis.* Bacterial strains were grown anaerobically in brain heart infusion (BHI) broth. For treatment, cultures were aliquoted onto a 96-well plate and treated with laser irradiation, anti-microbial at listed concentration, or combination therapy. Then, all samples were plated on trypticase soy agar, TSA blood agar plates post-treatment. After incubation, colonies were counted to judge the efficiency of treatment. Laser-treated samples were treated with an Er:YAG laser at the following settings: 40 mJ; 40 Hz; 1.6 W, 20 seconds, 300 µs short pulse duration, contact mode. Untreated cultures served as controls. Data is representative of four biological replicates. (*p* < 0.01=**, *p* < 0.001 = ***).

**Figure 2 microorganisms-07-00612-f002:**
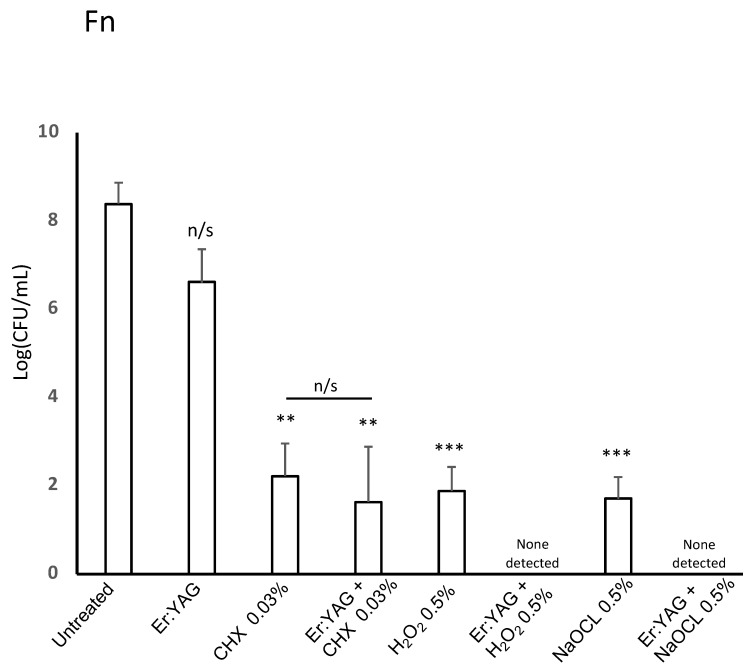
Effectiveness of Er:YAG laser treatment with or without irrigation on viability of *F. nucleatum.* Bacterial strains were grown anaerobically in BHI broth. For treatment, cultures were aliquoted onto a 96-well plate and treated with laser irradiation, anti-microbial at listed concentration, or combination therapy. Then, all samples were plated on TSA blood agar plates post-treatment. After incubation, colonies were counted to judge the efficiency of treatment. Laser-treated samples were treated with an Er:YAG laser at the following settings: 40 mJ; 40 Hz; 1.6 W, 20 seconds, 300 µs short pulse duration, contact mode. Untreated cultures served as controls. Data is representative of four biological replicates. (*p* < 0.01=**, *p* < 0.001 = ***).

**Figure 3 microorganisms-07-00612-f003:**
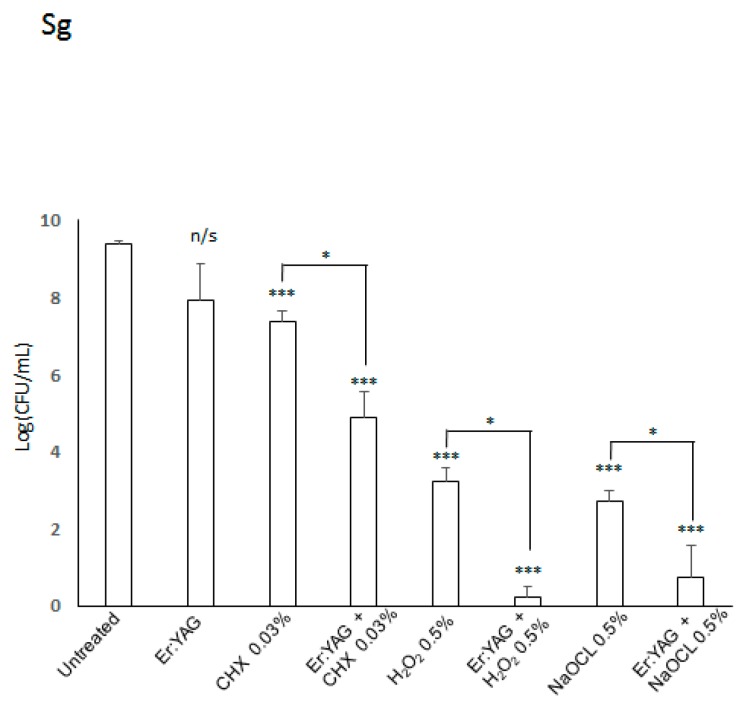
Effectiveness of Er:YAG laser treatment with or without irrigation on viability of *S. gordonii.* Bacterial strains were grown anaerobically in BHI broth. For treatment, cultures were aliquoted onto a 96-well plate and treated with laser irradiation, anti-microbial at listed concentration, or combination therapy. Then, all samples were plated on TSA blood agar plates post-treatment. After incubation, colonies were counted to judge the efficiency of treatment. Laser-treated samples were treated with an Er:YAG laser at the following settings: 40 mJ; 40 Hz; 1.6 W, 20 s, 300 µs short pulse duration, contact mode. Untreated cultures served as controls. Data is representative of four biological replicates. (*p* < 0.5 = *, *p* < 0.001 = ***).

**Figure 4 microorganisms-07-00612-f004:**
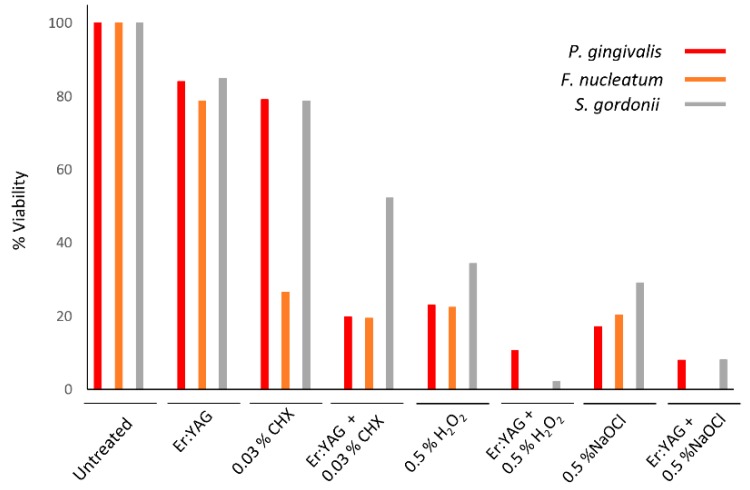
Comparison of all treatment modalities on the viability of the three bacterial species used in our studies. Data shown as percent viability with respect to the untreated control. All bacteria were grown anaerobically in BHI broth. For treatment, cultures were aliquoted onto a 96-well plate and treated with laser irradiation, anti-microbial at listed concentration, or combination therapy. Then, all samples were plated on TSA blood agar plates post-treatment. After incubation, colonies were counted to judge the efficiency of treatment. Laser-treated samples were treated with Er:YAG laser at the following settings: 40 mJ; 40 Hz; 1.6 W, 20 seconds, 300 µs short pulse duration, contact mode. Untreated cultures served as controls. Data is representative of four biological replicates.

**Table 1 microorganisms-07-00612-t001:** Colony counts for bacteria following various treatments. The number of colonies is shown as colony-forming units (CFU)/mL. *p*-value is presented for each condition with respect to the untreated control as well as between irradiated and non-irradiated samples using the same irrigation solution.

	Untreated	Er:YAG	CHX 0.03%	Er:YAG + CHX	H2O2 0.5%	Er:YAG + H2O2	NaOCl 0.5%	Er:YAG + NaOCl
***P. gingivalis***	9.5 ± 0.06	7.98 ± 1.75; *p* = 0.22	7.51 ± 0.29; *p* = 0.009	1.88 ± 1.8; *p* = 0.005	2.18 ± 0.48; *p* = 0002	1.0 ± 1.0; *p* = 0006	1.62 ± .53; *p* = 0001	0.75 ±0.83; *p* = 0003
			*p* = 0.009		*p* = 0.15		*p* = 0.15	
***F. nucleatum***	8.39 ± 0.48	6.6 ± 0.76; *p* = 0.062	2.22 ± 0.74; *p* = 0.003	1.63 ± 1.25; *p* = 0.003	1.88 ± 0.55; *p* = 0.0007	N.D.*	1.7 ± 0.49; *p* = 0.0009	N.D.*
			*p* = 0.47					
***S. gordonii***	9.37 ± 0.08	7.94 ± 0.92; *p* = 0.065	7.37 ± 0.26; *p* = 0.0002	4.89 ± 0.67; *p* = 0.0009	3.22 ± 0.37; *p* = 0.0001	0.2 ± 0.28; *p* = 0.0004	2.72 ± 0.27; *p* < 0.00001	0.75 ± 0.8; *p* = 0.0004
			*p* = 0.031		*p* = 0.024		*p* = 0.04	

*N.D. – not determined (no bacterial colonies detected).

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
