# Peer review of "Er:YAG Laser Irradiation Reduces Microbial Viability When Used in Combination with Irrigation with Sodium Hypochlorite, Chlorhexidine, and Hydrogen Peroxide"

_microorganisms, 2019, doi:10.3390/microorganisms7120612_

Round 1

Reviewer 1 Report

very interesting topic

smart results

Some additional specific comments on the manuscript. The combination of irrigants with and without the effect of the Er:YAG Laser is the only comparison that makes sense and the first paper of this kind I have seen. The stated reasons for the decrease of the concentrations of the rinsing solutions can be accepted. The results are of big interest because clinically we use much higher concentrations for NaOCl and H²O² and CHX. The mentioning of the limitations of this in- vitro-study are appreciated the results are clear that only the combination of Er:YAG laser and irrigants can cause effectiveness in the reduction of these cultures of bacterias, neither laser nor irrigants alone have these capabilities

Author Response

This is a response to suggested revision to Manuscript, Er:YAG Laser Irradiation Reduces Microbial Viability When Used in Combination with Irrigation with Sodium Hypochlorite, Chlorhexidine and Hydrogen Peroxide”(microorganisms-639132),

The authors would like to thank the reviewers for their thorough review and recommendations to improve the quality of the manuscript. We have carefully considered your suggestions and implemented the changes into a revised manuscript. In blue are the replies to the reviewers' comments outlining the changes made in the manuscript to address the questions/criticisms of the reviewers.

Reviewer: 1

Reviewer Comments to Author:
Some additional specific comments on the manuscript. The combination of irrigants with and without the effect of the Er:YAG Laser is the only comparison that makes sense and the first paper of this kind I have seen. The stated reasons for the decrease of the concentrations of the rinsing solutions can be accepted. The results are of big interest because clinically we use much higher concentrations for NaOCl and H²O² and CHX. The mentioning of the limitations of this in- vitro-study are appreciated the results are clear that only the combination of Er:YAG laser and irrigants can cause effectiveness in the reduction of these cultures of bacterias, neither laser nor irrigants alone have these capabilities 

The authors thank the reviewer for his/her comments with great appreciation for the feedback.

We have reviewed the English language and style and implemented minor spelling corrections.

Reviewer 2 Report

The authors describe the effect of laser irradiation and laser irradiation combined with exposure to chlorhexidine ,hydrogen peroxide or sodium hypochlorite on the viability of 3 different oral bacteria, e.g. Streptococcus gordonii , Fusobacterium nucleatum and Porphyromonas gingivalis. Overall the manuscript presents preliminary but interesting results. The drawbacks were that this study was only done with single species, and tested on cell suspensions, not biofilms that are the main form present in the oral cavity.

Italic is missing in all the species names and the in vitro. The first time species are mentioned in the manuscript, the full name should be stated. Streptococcus gordonii instead of S.gordonii

Abstract

Should be more concise. Material and method section is very detailed.

Introduction

Line 77 please state the beneficial biological effects on the host

Materials and Methods

In the control cells were the cells were moved to a chamber with no treatment, to check the effect of aerobiosis on growth?

Results

154 Figures 1 and 4? Should be 1 and 2

Table 1 and figure 1, 2 and 3 give the same information, that is also the same information presented in figure 4. Data should be presented only once.

In figure 4 the percentage of viability should be presented instead of CFUs. This way the initial CFU value would not interfere with the evaluation of the effect of each treatment.

Table 2 should be removed from the manuscript and included as supplementary data. The statistical significance should be included in the figures or table 1.

Author Response

This is a response to suggested revision to Manuscript, Er:YAG Laser Irradiation Reduces Microbial Viability When Used in Combination with Irrigation with Sodium Hypochlorite, Chlorhexidine and Hydrogen Peroxide”(microorganisms-639132),

The authors would like to thank the reviewers for their thorough review and recommendations to improve the quality of the manuscript. We have carefully considered your suggestions and implemented the changes into a revised manuscript. In blue are the replies to the reviewers' comments outlining the changes made in the manuscript to address the questions/criticisms of the reviewers.

Reviewer: 2

The authors describe the effect of laser irradiation and laser irradiation combined with exposure to chlorhexidine ,hydrogen peroxide or sodium hypochlorite on the viability of 3 different oral bacteria, e.g. Streptococcus gordonii , Fusobacterium nucleatum and Porphyromonas gingivalis. Overall the manuscript presents preliminary but interesting results. The drawbacks were that this study was only done with single species, and tested on cell suspensions, not biofilms that are the main form present in the oral cavity.

The authors agree with the comment and will design future experiments to study the effects of bacterial biofilm.

We added a sentence to discussion to address this concern, Line 287

Future studies should be designed to explore if these therapies have the same effects on a more complex microbial biofilm as they do on individual bacteria. Potential interactions among different bacteria and use of a more complex biofilm would bring a model closer to oral cavity.

Italic is missing in all the species names and the in vitro. The first time species are mentioned in the manuscript, the full name should be stated. Streptococcus gordonii instead of S.gordonii

Thank you for pointing these inconsistencies out to us. The authors have corrected the text format of all bacterial names into the italic format as well as used the full names of bacterial species when first mentioned in the introduction.

Abstract

Should be more concise. Material and method section is very detailed.

We shortened the Materials and Methods section in the abstract.

Introduction

Line 77 please state the beneficial biological effects on the host

The authors have listed the beneficial biological effects to the host to the existing sentence.

The more detailed description of biologic effects of the laser are also outlined in the discussion:

“Human histological studies have shown that treatment with Er:YAG laser results in favorable healing, formation of new attachment and may result in radiographic bone regeneration.[37] “

Irradiation with the Er:YAG laser facilitates root debridement and has positive effects on tissue repair resulting in greater probing depth reduction and gains in clinical attachment level that are sustained for up to 3 years compared to flap surgery.[39]

Hard-tissue lasers such as Er:YAG are the most promising and widely employed for root-surface and bone-defect debridement during surgery[13,22,38] with capability of extensive bacterial eradication into deeper layers of dentin at no significant temperature increase.[48] More complex biological effects include biostimulation, which may help accelerate wound healing and tissue regeneration.[16,17,50]

Materials and Methods

In the control cells were the cells were moved to a chamber with no treatment, to check the effect of aerobiosis on growth?

During our experiments we took care to not expose the anaerobic bacteria to oxygen more than necessary. The control bacterial samples were exposed to aerobic environment for the same amount of time as all samples (~10 minutes). This short time has a negligible effect on bacteria viability. Furthermore, immediately after treatment the bacteria were diluted 1:10 in fresh anaerobic medium as an extra pre-caution.

The authors have added a sentence in paragraph about culture conditions to address this concern, Line 115.

Results

154 Figures 1 and 4? Should be 1 and 2

Pg result are presented in Figure 1 (only Pg) and Figure 4 (combined figure for all three bacteria), therefore the figures are listed correctly.

Table 1 and figure 1, 2 and 3 give the same information, that is also the same information presented in figure 4. Data should be presented only once.

The authors have removed Table 2.

The authors would prefer to keep all 4 figures. The Figures 1-3 show each bacteria individually as well as the standard deviations on bars. Figure 4 enables a good overview and overall comparison of all treatment modalities on the three bacteria, which would be difficult to extrapolate form three separate figure.

Table 1 is valuable addition to demonstrate the statistical significance.

 In figure 4 the percentage of viability should be presented instead of CFUs. This way the initial CFU value would not interfere with the evaluation of the effect of each treatment.

As recommended, we have adjusted figure 4. It now shows the % viability of each bacteria based on the treatments.

Table 2 should be removed from the manuscript and included as supplementary data. The statistical significance should be included in the figures or table 1.

The authors have removed Table 2. The statistical significance of each condition is present in each figure (P<0.05=*, P<0.01=**, P<0.001=***) and the P-values have been added to table 1.